# PRACTICAL KERNEL LEARNING FOR KERNEL-BASED CONDITIONAL INDEPENDENCE TEST

## ABSTRACT

Conditional independence (CI) test stands as a fundamental and challenging task within modern statistics and machine learning. One pivotal class of methods for assessing conditional independence encompasses kernel-based approaches, known for their capability to identify general conditional dependence without necessitating assumptions about the conditional relationship or resorting to the simulation of intricate conditional distributions. As with any method utilizing kernels, selecting the appropriate kernel in kernel-based CI methods is critical for ensuring heightened test power and precise identification of conditional relationship. However, current methods typically involve the manual heuristic selection of kernel parameters, neglecting the inherent characteristics of the data and potentially leading to errors. In this paper, we propose a kernel parameter selection approach for the Kernel-based Conditional Independence test (KCI). We decompose the statistic of KCI and treat the kernel applied on the conditioning set as a trainable component. The kernel parameters involved are then learned by maximizing the ratio of the estimated statistic to its variance, which approximates the test power at large sample sizes. Therefore, our method can learn the kernel parameters with increased test power at a very small additional computation cost. Extensive experiments demonstrate the effectiveness of our proposed approach in conditional independence testing and its enhancements to constraint-based causal discovery.

## 1 INTRODUCTION

Conditional independence (CI) test is a cornerstone of statistics and machine learning. Let $X, Y$ and $Z$ denote sets of random variables, then the conditional independence relationship between $X$ and $Y$ given $Z$, denoted by $X \perp\!\!\!\perp Y \mid Z$, indicates that knowing the values of $Z$, the knowledge of $X$ does not yield any extra information about $Y$. This conditional independence relationship enables the removal of redundant variables when constructing probabilistic models for a given variable set. Therefore, the utilization of CI has expanded across diverse domains, including causal discovery (Spirtes et al., 2000; 1995; Pearl et al., 2000; Huang et al., 2020), fairness representation learning (Mehrabi et al., 2021), feature selection (Fukumizu et al., 2009; Song et al., 2012) and other machine learning areas (Long et al., 2018; Pogodin et al., 2022).

Traditional CI testing methods either address the discrete case or rely on simplifying assumptions to handle the continuous case. The discrete approach requires a substantial amount of data to comprehensively evaluate each potential configuration of the conditioning set $Z$ (Margaritis, 2005; Huang, 2010). Meanwhile, methods handling the continuous case often impose strong assumptions on the relationships between variables, such as linear associations with additive Gaussian errors (Lawrance, 1976) or other specific forms of nonlinear functions (Linton & Gozalo, 1996; Song, 2009). These assumptions can be restrictive, and when violated or when data is limited—conditions frequently encountered in practical applications—these methods often yield biased estimates and erroneous inferences, resulting in unreliable conclusions.

Daudin (1980) extended the concept of partial correlation to general scenarios involving nonlinear and non-Gaussian noise, redefining conditional independence as the zero correlation of any regression residual functions within constrained $L^2$ spaces. While this definition can identify general CI relationships, it requires considering all possible functions within these constrained $L^2$ spaces, which is infeasible. To make it practical, Zhang et al. (2011) relaxed the function spaces to reproduc-

ing kernel Hilbert spaces (RKHS) using kernel methods, simplifying computation while preserving the ability to capture general CI relationships. They introduced the Kernel-based Conditional Independence (KCI) statistic, which replaces the regression residuals with their kernel analogues. By employing characteristic kernels (Fukumizu et al., 2007), maximum mean discrepancy (MMD)-based statistics (Gretton et al., 2012a) can effectively measure distribution homogeneity. For CI testing, kernel methods enable direct assessment of whether $P_{XY|Z}P_Z$ equals $P_{X|Z}P_{Y|Z}P_Z$, thus bypassing the need to approximate complex conditional marginals such as $P_{X|Z}$ or $P_{Y|Z}$. Zhang et al. (2011) leverage conditional mean embedding (CME) (Song et al., 2009; Grünewälder et al., 2012) to model nonlinear relationships and replace the original cross-covariance defined in $L^2$ space with the Hilbert-Schmidt norm of the cross-covariance in RKHS (Gretton et al., 2005b) as the KCI statistic to detect correlations between residuals. Due to the reproducing property of kernel, a zero value of this statistic is equivalent to the partial correlations of any residual functions represented by the chosen kernels also being zero. Consequently, by employing characteristic kernels, whose RKHS are dense in $L^2$ (Sriperumbudur et al., 2008), KCI can identify general forms of conditional dependence, surpassing the detection of mere linear correlations. However, as with all kernel-based methods, the performance of KCI is directly influenced by the choice of kernels.

It is well-known that the effectiveness of kernel-based methods critically depends on the selection of appropriate kernels (Brockmann et al., 1993; Chapelle & Vapnik, 1999), making kernel selection a significant challenge across various tasks involving kernel methods. This selection process primarily focuses on tuning kernel parameters, such as the bandwidth in the radial basis function (RBF) kernel, which can often be more influential than the choice of the kernel family itself (Schölkopf et al., 2002, Section 4.4.5). A commonly employed approach for kernel parameter selection is the median heuristic, where the kernel bandwidth is set to the median of the pairwise distances between data instances. Despite its widespread use, this simple heuristic may not always be the most suitable for the data at hand (Ramdas et al., 2015; Garreau et al., 2017). Therefore, the ability to find better kernel parameters based on the given data is crucial for the performance of all kernel-based methods. Although various methods for kernel bandwidth selection have been proposed to address this goal (Sriperumbudur et al., 2008; Gretton et al., 2012b; Sutherland et al., 2021), CI testing requires indirectly considering the correlations between residuals rather than the original data, which is significantly different from other hypothesis testing tasks. Therefore, suitable kernel selection methods are still lacking for CI testing.

**Contributions.** In this paper, we propose a kernel selection method to optimize the kernel parameters involved in the widely used KCI statistic. Given the unique characteristics of CI test, which necessitates indirect consideration of regression residuals that inherently contain regression bias, we first decompose the original KCI statistic to isolate the kernel component associated with the conditioning set, which was previously mixed within the residuals. We treat the parameters associated with this kernel as trainable, while keeping other parameters fixed to avoid introducing additional regression bias. These kernel parameters are then optimized to maximize the ratio of the estimated statistic to its variance, effectively maximizing the test power at large-sample size. Consequently, our method enhances kernel parameter selection, achieving higher test power and improved performance with minimal additional computational cost. The extensive experiments, including extensions to causal discovery tasks, demonstrate that our method consistently outperforms the median heuristic-based one in most scenarios. With the negligible computational overhead, our method shows promise as a replacement for the original median heuristic-based KCI statistic in a broad range of CI-related applications.

## 2 PRELIMINARIES

### 2.1 CONDITIONAL INDEPENDENCE TESTING

Suppose there are three random variables $X$, $Y$ and $Z$ with observational points, and their joint distribution is absolutely continuous with respect to Lebesgue measure with density $P$. The problem of testing CI between $X$ and $Y$ given $Z$ can be written in the form of a hypothesis testing:

$$\text{H}_0 : X \perp\!\!\!\perp Y \mid Z \quad \text{versus} \quad \text{H}_1 : X \not\perp\!\!\!\perp Y \mid Z.$$

CI testing generally consists of the following procedure: define a statistic $T$ and select a significance level $\alpha \in [0, 1]$ (typically set at 0.05); compute the test statistic value $\hat{T}$ from the observational data;

compute the $p$-value, which is the probability of returning a statistic as large as $\hat{T}$ when $\mathrm{H}_0$ is true; finally, reject $\mathrm{H}_0$ if the $p$-value is not greater than $\alpha$. There are two types of errors may in hypothesis testing: type I error is a probability of rejecting $\mathrm{H}_0$ when it actually holds, and Type II error is a probability of failing to reject $\mathrm{H}_0$ when $\mathrm{H}_1$ holds. A well performed CI test requires Type I error rate not greater than the chosen significance level while making Type II error as low as possible.

Due to the unique nature of CI testing, Shah & Peters (2020) demonstrated that a valid CI test does not have power against any alternatives. This implies that no method can simultaneously control the Type I error rate at the given significance level while maintaining adequate power. Consequently, the practical evaluation of CI methods necessitates a balanced assessment of both Type I and Type II error rates, emphasizing the trade-off between error control and statistical power.

## 2.2 RELATED WORK

There is a growing body of literature on conditional independence test, which can be roughly divided into three groups: (1) regression-based methods (Shah & Peters, 2020; Scheidegger et al., 2022; He et al., 2021; Polo et al., 2023); (2) simulation-based methods (Doran et al., 2014; Candes et al., 2018; Berrett et al., 2020) and (3) kernel-based methods (Fukumizu et al., 2007; Zhang et al., 2011; Kour & Saabne, 2014). Regression-based methods require assumptions about the relationship and noise structure, as well as the assumptions of removal of any information from the conditioning set $Z$ by regression. When these assumptions hold, regression-based methods have been shown to effectively control Type I error; otherwise, they do not. Another important category is simulation-based methods (also known as randomization-based methods), which primarily implicitly or explicitly approximate the conditional distributions $P_{X|Z}$ or $P_{Y|Z}$ to simulate the null distribution. A clear drawback is that such approaches often come with significant approximation errors, leading to an inflation of the type-I error and rendering the test invalid.

Kernel-based CI methods, on the other hand, do not require additional assumptions and can detect general dependence. By mapping variables into a RKHS, kernel functions enable the assessment of similarities between high-dimensional implicit functions, thereby capturing higher-order statistical moments. Utilizing characteristic kernels allows us to infer distribution properties such as homogeneity (Gretton et al., 2012a), independence (Gretton et al., 2005a), and conditional independence (Fukumizu et al., 2007; Sun et al., 2007; Zhang et al., 2011; Huang et al., 2022). These properties make kernel-based methods capable of discerning conditional independence in CI tasks without the need to simulate intricate conditional distributions.

In kernel-based methods, a critical aspect to consider is the choice of kernel functions, as they can directly affect the accuracy of the final results. The selection of appropriate kernels remains an unresolved question in numerous studies (Chu & Marron, 1991; Herrmann et al., 1992; Chapelle & Vapnik, 1999; Kim et al., 2006). Most existing works on kernel selection focus on homogeneity tasks, such as the two-sample test (Gretton et al., 2012b; Liu et al., 2020). Fukumizu et al. (2009) propose simply maximizing the MMD statistic itself, which is proven to be equivalent to minimizing the classification error under linear loss. However, it is not optimal due to the ignored variance component (Gretton et al., 2012b). For CI task, it has its own characteristics, primarily involving the consideration of regression residuals, which inherently contain biases. In this paper, we investigate the kernel selection for KCI (Zhang et al., 2011).

## 2.3 KERNEL-BASED MEASURES OF CONDITIONAL DEPENDENCE

We first provide the general characterization of conditional independence from the perspective of partial association.

**Definition 1.** (Daudin, 1980) Random variables $X$ and $Y$ are independent conditioned on $Z$, denoted $X \perp\!\!\!\perp Y \mid Z$, if for all functions $g \in L^2_{XZ}$ and $h \in L^2_Y$, we have almost surely in $Z$ that

$$\mathbb{E}[g(X, Z)\, h(Y) \mid Z] = \mathbb{E}[g(X, Z) \mid Z]\mathbb{E}[h(Y) \mid Z].$$

**Theorem 2.** (Daudin, 1980) $X \perp\!\!\!\perp Y \mid Z$ if and only if

$$\mathbb{E}[g(X, Z)h(Y)] = 0 \quad \forall g \in E_1, h \in E_2, \tag{1}$$

where $E_1 = \{g \in L^2_{XZ} : \mathbb{E}[g(X, Z) \mid Z] = 0\}$ and $E_2 = \{h \in L^2_Y : \mathbb{E}[h(Y) \mid Z] = 0\}$.

Since $g(X, Z)$ can represent any general relationship between $X$ and $Z$, Theorem 2 can be intuitively understood as asserting that the residuals obtained from regressing any function mappings of $(X, Z)$ and $Y$, defined in the $L^2$ space, onto $Z$ are uncorrelated. Therefore, this definition can capture general CI relationships but requires considering all possible functions in $L^2$.

To use this characterization in practice, Zhang et al. (2011) introduce it within the RKHS. For the random variable $X$ with its domain $\mathcal{X}$, we define the RKHS $\mathcal{H}_{\mathcal{X}}$ on $\mathcal{X}$ with a symmetric positive-definite function $k_{\mathcal{X}} : \mathcal{X} \times \mathcal{X} \to \mathbb{R}$. The kernel can be represented as an inner product in $\mathcal{H}_{\mathcal{X}}$ via a mapping $\phi_x : \mathcal{X} \to \mathcal{H}_{\mathcal{X}}$, which is $k_{\mathcal{X}}(x, x') = \langle \phi_x(x), \phi_x(x') \rangle$. And with the reproducing property, we have $\forall x \in \mathcal{X}$ and $\forall f \in \mathcal{H}_{\mathcal{X}}, f(x) = \langle f, \phi_x(x) \rangle$. Similar to the notation on $X$, we define $(k_{\mathcal{Y}}, \phi_y(Y), \mathcal{H}_{\mathcal{Y}}), (k_{\mathcal{Z}}, \phi_z(Z), \mathcal{H}_{\mathcal{Z}})$ and $(k_{\mathcal{X}\mathcal{Z}}, \phi_{xz}(X, Z), \mathcal{H}_{\mathcal{X}\mathcal{Z}})$ with $k_{\mathcal{X}\mathcal{Z}} \coloneqq k_{\mathcal{X}} k_{\mathcal{Z}}$ . Building upon the cross-covariance operator (Fukumizu et al., 2007), Zhang et al. (2011) then propose the Kernel-based Conditional Independence (KCI) statistic for CI testing, which is defined as follows:

$$\Sigma_{\ddot{X}Y|Z} = \mathbb{E}[(\phi_{xz}(X, Z) - \mu_{XZ|Z}(Z)) \otimes (\phi_y(Y) - \mu_{Y|Z}(Z))], \tag{2}$$

where $\ddot{X}$ represents $(X, Z)$, $\otimes$ is the tensor product, $\mu_{XZ|Z}$ and $\mu_{Y|Z}$ represent the conditional mean embeddings given by $\mu_{XZ|Z}(Z) = \mathbb{E}[\phi_{xz}(X, Z) \mid Z]$ and $\mu_{Y|Z}(Z) = \mathbb{E}[\phi_y(Y) \mid Z]$. Utilizing the property that for any $g \in \mathcal{H}_{\mathcal{X}\mathcal{Z}}$ and $h \in \mathcal{H}_{\mathcal{Y}}$ (see e.g. Gretton (2013, Lecture 5)), the tensor product operates as $(\phi_{xz} \otimes \phi_y)g = \langle \phi_{xz}, g \rangle \phi_y$, we can derive the following equation:

$$\left\langle h, \Sigma_{\ddot{X}Y|Z}g \right\rangle = \mathbb{E}[(g(X, Z) - \mathbb{E}[g(X, Z) \mid Z])(h(Y) - \mathbb{E}[h(Y) \mid Z])],$$

which holds for any $g \in \mathcal{H}_{\mathcal{X}\mathcal{Z}}$ and $h \in \mathcal{H}_{\mathcal{Y}}$. For a class of kernel functions known as characteristic kernels (such as Gaussian kernel), their RKHSs are dense in $L^2$ spaces (Sriperumbudur et al., 2008). With characteristic kernels employed, if $\Sigma_{\ddot{X}Y|Z} = 0$, Eq. 1 holds for any $g \in E_1 \cap \mathcal{H}_{\mathcal{X}\mathcal{Z}}$ and $h \in E_2 \cap \mathcal{H}_{\mathcal{Y}}$, encompassing sufficient functions by continuity and density. This implies that $\Sigma_{\ddot{X}Y|Z} = 0$ if and only if $X \perp\!\!\!\perp Y \mid Z$. Therefore, we can test conditional independence by evaluating whether the Hilbert-Schmidt norm of the operator is zero, i.e. $\|\Sigma_{\ddot{X}Y|Z}\|_{\mathrm{HS}}^2 = 0$.

## 3 POWER-BASED KERNEL LEARNING FOR CONDITIONAL INDEPENDENCE TESTING

In all kernel-involved methods, the choice of kernel parameters is crucial, and KCI is no exception. The kernel parameters involved in KCI directly influence its performance, as they play a crucial role in more effectively controlling the Type I error at the specified significance level and achieving higher test power with reduced Type II error. However, like most kernel-based approaches, KCI relies on the median heuristic to determine its kernel parameters. While this setup is straightforward, it may not fully capture the inherent characteristics of the data, potentially leading to inaccurate assessments of the CI relationship. In this paper, we propose a power-based kernel selection method for the kernels involved in KCI, named **Power**, aiming to enhance its performance in CI tasks.

**Decomposition of KCI.** We first decompose the kernel mapping of the conditioning set $Z$ from the concatenated $\phi_{zx}(X, Z)$ in the original form (i.e. Eq. 1). According to (Mastouri et al., 2021; Pogodin et al., 2022), the RBF kernels (e.g. Gaussian and Laplace kernel) of $\phi_{zx}(X, Z)$ can be decomposed into $\phi_x(X) \otimes \phi_z(Z)$. For the conditional expectation, we can derive that $\mu_{XZ|Z}(Z) = \mathbb{E}[\phi_x(X) \otimes \phi_z(Z) \mid Z] = \mathbb{E}[\phi_x(X) \mid Z] \otimes \phi_z(Z)$. Then, we derive the decomposed form of the KCI statistic, which isolates $\phi_z(Z)$ from the regression residual of $\phi_{xz}(X, Z)$ with respect to $Z$:

$$\Sigma_{\ddot{X}Y|Z} = \mathbb{E}[\phi_z(Z) \otimes (\phi_x(X) - \mu_{X|Z}(Z)) \otimes (\phi_y(Y) - \mu_{Y|Z}(Z))]. \tag{3}$$

**Benefits of the Decomposition.** The decomposition avoids estimating the identity operator $\mu_{Z|Z} = \phi(Z)$, which is not Hilbert-Schmidt in characteristic RKHS (Mastouri et al., 2021), leading to an ill-specified regression problem with biased estimates in the tail due to data scarcity[1]. On the other hand, isolating $\phi_z$ from $\phi_{zx}$ allows for the direct optimization of $\phi_z$ without being influenced by the estimation bias of $\mu_{X|Z}$. In other words, the presence of conditional expectation bias makes it challenging to obtain the expected residuals with higher test power when updating the kernel

---

[1]We empirically analyzed this decomposition in the Ablation study; see Section 4.1.2 for further discussion.

applied to the regressed variables (i.e. $\phi_x$ and $\phi_y$), whereas $\phi_z$ remains unaffected by these biases. Therefore, we propose to selectively learn the parameters involved in $\phi_z$ to achieve better test power.

**Asymptotic Normality.** We now describe an empirical estimate of $\|\Sigma_{\ddot{X}Y|Z}\|^2_{\text{HS}}$. For simple notation, we denote $\|\Sigma_{\ddot{X}Y|Z}\|^2_{\text{HS}}$ as $\text{C}^2_{\text{KCI}}$. Then, we express $\text{C}^2_{\text{KCI}}$ as follows:

$$\text{C}^2_{\text{KCI}} = \mathbb{E}\left[k_{\mathcal{Z}}(z, z')\left\langle\phi_{x|z}(z), \phi_{x|z}(z')\right\rangle\left\langle\phi_{y|z}(z), \phi_{y|z}(z')\right\rangle\right], \tag{4}$$

where $z$ and $z'$ are independent copies of $Z$, $k_{\mathcal{Z}}(\cdot, \cdot)$ is the kernel function associated with $\mathcal{H}_{\mathcal{Z}}$ defined by $k_{\mathcal{Z}}(z, z') = \langle\phi_z(z), \phi_z(z')\rangle$, and $\phi_{x|z}(z)$ is the regression residual $\phi_{x|z}(z) = \phi_x(x) - \mu_{X|Z}(z)$. Suppose we have $n$ i.i.d. observational points $S = \{s_i\}_{i=1}^n$ with $s_i = (x_i, y_i, z_i)$ being the one sample pair of $(X, Y, Z)$. We can intuitively give an unbiased U-statistic estimator for $\text{C}^2_{\text{KCI}}$, given by:

$$\widehat{\text{C}}^2_{\text{KCIu}} = (n)_2^{-1}\sum_{i,j\neq i} h(i, j), \tag{5}$$

where $h(i, j) = K_{Z(i,j)}K_{X|Z(i,j)}K_{Y|Z(i,j)}$ with $K_{Z(i,j)} = k_{\mathcal{Z}}(z_i, z_j)$ and the residual $K_{X|Z(i,j)} = \left\langle\phi_x(x_i) - \mu_{X|Z}(z_i), \phi_x(x_j) - \mu_{X|Z}(z_j)\right\rangle$ (and similarly for $K_{Y|Z}$). The U-statistic $\widehat{\text{C}}^2_{\text{KCIu}}$ has expectation zero under the null hypothesis $\text{H}_0$ that $X \perp\!\!\!\perp Y \mid Z$, and has a strictly positive expected value under the alternative $\text{H}_1$ that $X \not\perp\!\!\!\perp Y \mid Z$.

$\widehat{\text{C}}^2_{\text{KCIu}}$ is the straightforward average of independent random variables, and its asymptotic distribution is given by the central limit theorem (see e.g. Lee (2019, Section 3.2.1)). If $\mathbb{E}(h^2) < \infty$ (which is true for bounded continuous kernels), then under the alternative $\text{H}_1$ where $X \not\perp\!\!\!\perp Y \mid Z$, we have:

$$\sqrt{n}\left(\widehat{\text{C}}^2_{\text{KCIu}} - \text{C}^2_{\text{KCI}}\right) \xrightarrow{d} \mathcal{N}(0, 4\sigma_1^2), \tag{6}$$

where $\sigma_1^2$ is the asymptotic variance, which is given by $\sigma_1^2 = \text{Var}[h_1(s_i)]$ with $h_1(s_i) = \mathbb{E}_{s_j\neq s_i}[K_{Z(i,j)}K_{X|Z(i,j)}K_{Y|Z(i,j)}]$. With the fact that $\mathbb{E}_{s_i}[h_1(s_i)] = \text{C}^2_{\text{KCI}}$, we can derive that

$$\sigma_1^2 = \text{Var}[h_1(s_i)] = \mathbb{E}_{s_i}[(h_1(s_i) - \mathbb{E}_{s_i}[h_1(s_i)])^2] = \mathbb{E}_{s_i}[\mathbb{E}_{s_j}[h(s_i, s_j)] - \text{C}^2_{\text{KCI}}]^2. \tag{7}$$

**Test Power.** Based on the asymptotic normality in Eq. 6, we can estimate the test power, which represents the probability of correctly rejecting $\text{H}_0$ when $\text{H}_1$ is true for a given case. Assuming that the conditional expectations are well estimated, the power of our test is thus, using $\text{Pr}_1$ to denote the probability under $\text{H}_1$,

$$\text{Pr}_1\left(n\widehat{\text{C}}^2_{\text{KCIu}} > r\right) = \text{Pr}_1\left(\frac{n(\widehat{\text{C}}^2_{\text{KCIu}} - \text{C}^2_{\text{KCI}})}{2\sqrt{n}\sigma_1} > \frac{r - n\text{C}^2_{\text{KCI}}}{2\sqrt{n}\sigma_1}\right)$$

$$\to \Phi\left(\frac{\sqrt{n}\text{C}^2_{\text{KCI}}}{2\sigma_1} - \frac{r}{2\sqrt{n}\cdot\sigma_1}\right),$$

where $\Phi$ is the CDF of the standard normal distribution and $r$ is the rejection threshold, which is a constant for a specified significance level. The test power therefore can be maximized by maximizing the argument in $\Phi$. Since the statistic $\text{C}^2_{\text{KCI}}$ and the asymptotic variance $\sigma_1$ are also constant, for reasonable large sample size $n$, the power will be dominated by the first term, i.e. $\sqrt{n}\text{C}^2_{\text{KCI}}/2\sigma_1$. So following (Sutherland et al., 2021; Liu et al., 2020), we can asymptotically maximize the test power by learning the kenrel parameters that increases the ratio of $\text{C}^2_{\text{KCI}}$ to $\sigma_1$.

**Learning kernels.** Both $\text{C}^2_{\text{KCI}}$ and $\sigma_1$ depend on the distribution at hand, making them unable to be estimated with finite samples. In practice, we use their empirical estimators from training samples. That is, we learn the kernel parameters involved, denoted as $\boldsymbol{\theta}_k$, to maximize

$$\hat{J}(S, \boldsymbol{\theta}_k) = \widehat{\text{C}}^2_{\text{KCIu}}/\widehat{\sigma}_1, \tag{8}$$

where $\widehat{\sigma}_1$ is the estimated asymptotic variance from finite sample averages:

$$\widehat{\sigma}_1^2 = \frac{1}{n}\sum_i[h_1(s_i) - \widehat{\text{C}}^2_{\text{KCIu}}]^2 = \frac{1}{n}\sum_i\left[\left(\frac{1}{n-1}\sum_{j\neq i}h(i,j)\right) - \widehat{\text{C}}^2_{\text{KCIu}}\right]^2, \tag{9}$$

where $h(i, j) = K_{Z(i,j)}K_{X|Z(i,j)}K_{Y|Z(i,j)}$. Thus, we opt to learn the kernel parameters involved in $\phi_z$ to maximize $\hat{J}(S, \boldsymbol{\theta}_k)$, and then use the learned kernels to conduct the final hypothesis test.

**Overall test procedure.** The full testing procedure of our Power method involves several steps. (1) Choose characteristic kernels and determine the parameters for $\phi_x$ and $\phi_y$. The kernel parameters involved in $\phi_x$ and $\phi_y$ remain fixed throughout the entire procedure. (2) Using kernel ridge regression (Bach & Jordan, 2002) to estimate $\mu_{X|Z}$ and $\mu_{Y|Z}$, denoted as $\widehat{\mu}_{X|Z} = K_Z^R (K_Z^R + \varepsilon I)^{-1} \phi_x(X)$, where $\varepsilon$ is the trainable regularization parameter and $K_Z^R$ represents the kernel matrix of $Z$ in the kernel ridge regression. (3) Obtaining the residual matrix $K_{X|Z}$ based on the estimated regression $\widehat{\mu}_{X|Z}$, where $K_{X|Z} = R_Z K_X R_Z$ with $R_Z = \varepsilon(K_Z^R + \varepsilon I)^{-1}$ and $K_{X(i,j)} = k_{\mathcal{X}}(x_i, x_j)$. Similarly, obtain $K_{Y|Z}$ in the same manner. (4) Choose characteristic kernels and initialize the kernel parameter for $\phi_z$ and learn the kernel parameters of $\phi_z$ by maximizing Eq. 8. (5) Compute the HSIC-like estimator (Gretton et al., 2005a) on testing point as follows,

$$\widehat{C}_{\text{KCIb}}^2 = \frac{1}{n(n-1)} \text{Tr}(H K_{X|Z} H (K_Z \odot K_{Y|Z})), \tag{10}$$

where $n$ is the number of test samples, $H = I - \dfrac{1}{n} 1_n 1_n^\top$ is the centering matrix with $I$ and $1_n$ being the $n \times n$ identity matrix and the vector of 1's, respectively. (6) Approximate the null distribution and compute the $p$-value (See Appendix A.1 for more details).

**On the choice of learnable kernels.** In Power, we only optimize the kernel parameters of $\phi_z$ while keeping $\phi_x$ and $\phi_y$ fixed during the training procedure. Theoretically, the kernel parameters of $\phi_x$ and $\phi_y$ can also be optimized using our proposed criteria if the conditional mean embedding $\mu_{X|Z}$ and $\mu_{Y|Z}$ can be well estimated without bias. However, in practice, due to the presence of conditional expectation bias, we empirically found that updating these parameters does not yield residual matrices with higher expected test power, as we will illustrate shortly. Therefore, we choose to optimize only the kernel parameters of $\phi_z$, taking a step towards higher test power even in the presence of existing bias.

## 4 EXPERIMENTAL RESULTS

In this section, we employ our proposed Power method to conduct CI tests on both synthetic and real benchmark, evaluating its empirical performance across various scenarios and comparing it with the median heuristic-based method and other baseline approaches. Additionally, we apply this method to the causal discovery task, using a search algorithm to assess its improvements in this context.

**Implementation details.** We use Gaussian kernels for all the kernels involved. For the kernels in $\phi_x$ and $\phi_y$, we set their bandwidth using the median heuristic, which is twice the median distance between the input points in the original data space. For the kernel parameters involved in $K_Z^R$ from the kernel ridge regression, we also use the median heuristic to initialize the bandwidth and employ a Gaussian process to train these parameters along with other parameters in the model. Additionally, we use the median heuristic to initialize the bandwidth of $k_{\mathcal{Z}}$ related to $\phi_z$ in the statistic and optimize it using Adam (Kingma & Ba, 2014) as the optimization algorithm. During the testing phase, we use the weighted sum of chi-squared to compute $p$-value. Please refer to Appendix B.1 for more implementation details.

### 4.1 SYNTHETIC DATA

In the synthetic experiment, we assume $X$ and $Y$ are dependent variables conditioned on $Z$. We analyzed our method's performance under varying dimensions of the conditioning set $Z$ and different sample sizes. To clearly examine Type I errors in scenarios where $X$ and $Y$ should be independent given $Z$, we generated $X$ and $Y$ using the following post-nonlinear functional model:

$$X = g\left(\sum_i f_i(Z_i) + E\right), \tag{11}$$

where $f_i$ and $g$ were randomly chosen from the *linear*, *sin*, *cos*, *tanh* and power function. The noise term $E$ was randomly chosen from either a *Gaussian* or *uniform* distribution. The number of $f_i$ is the same as the dimension of $Z$, with each $f_i$ being independently sampled. Therefore, the relationships between $Z$ and the dependent variable $X$ and $Y$ become more complex as the dimension of $Z$ increases. To examine Type II errors, we added an additional variable $T$ to both $X$ and $Y$, making

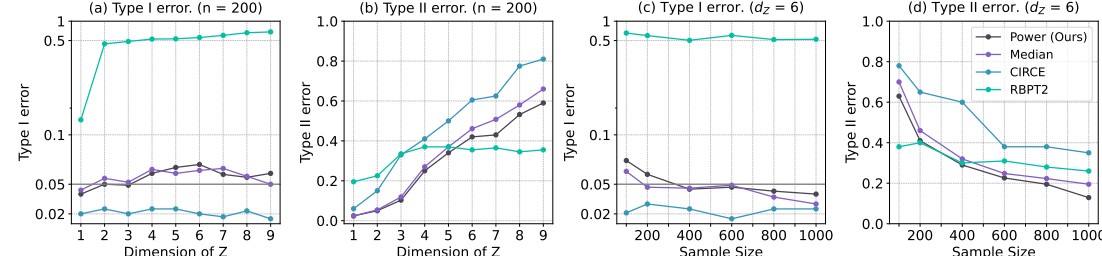

Figure 1: Performance on synthetic data with the significance level $\alpha = 0.05$ (gray line). Left: Type I error (a) and Type II error (b) when increasing the dimension of conditioning variable $Z$, keeping sample size $n = 200$. Right: Type I error (c) and Type II error (d) when increasing the number of samples, keeping the dimension $d_Z = 6$.

them conditionally dependent given $Z$. $T$ was sampled from a Gaussian distribution, and formally, $X/Y = g(\sum_i f_i(Z_i) + E) + T$. For each setting, we randomly repeated the process 1000 times to obtain Type I and Type II error. For further implementation details, please refer to Appendix B.2.

### 4.1.1 COMPARISON WITH BASELINE METHODS

**Baseline Models.** We first compare our proposed power-based method with CI baselines. Our proposed method is denoted as *Power*, while the median heuristic-based method is denoted as *Median*. In Median, the kernel bandwidth $\sigma_z$ involved in $K_Z$ is determined by the median heuristic and remains fixed throughout the entire procedure. In Power, we initialized the kernel bandwidth using the median heuristic and then optimized it based on the estimated power (i.e., Eq. 8), while keeping all other settings the same as those in Median. We further compare it with the kernel-based CIRCE (Pogodin et al., 2022), which only considers the independence between one-sided residuals and the other dependent variable itself. Additionally, we compare with regression-based method RBPT2 (Polo et al., 2023), which conduct the regression in $L^2$ space. (See Appendix A.2 for more details about CIRCE and RBPT2).

**On the dimension of $Z$.** Figure 1(a) and (b) illustrate the performance of Power and baseline methods, with a fixed sample size $n = 200$ and an increasing the dimension of $Z$ from 1 to 9. In general, all methods show varying biases in controlling Type I error, and Type II error generally rises with increasing $Z$. The regression-based RBPT2 struggles with controlling Type I error. The kernel-based CIRCE maintains a lower Type I error at $\alpha = 0.05$, but at the cost of higher Type II error. In contrast, Power and Median, based on bilateral regression residuals, demonstrate higher test power, with their Type I errors slightly exceeding the significance level as $Z$ increases. Notably, Power achieves slightly lower Type I and Type II errors compared to Median when $d_Z < 5$. As $d_Z$ grows, Power maintains a significantly lower Type II error than Median. Overall, Power consistently outperforms Median, demonstrating higher test power across different dimensions of $Z$, especially when the dimensions of $Z$ is higher. (For $n = 500$ see Figure 4 in the Appendix)

**On the sample size.** We also assessed the performance via varying sample sizes, shown in Figure 1(c) and (d). In general, all methods exhibit a reduction in Type II error as the sample size increases. The Type I error of RBPT2 remains disproportionately largely, while CIRCE consistently maintains a Type I error significantly below the significance level. Both Median and Power perform better across different sample sizes, demonstrating improved control of Type I error and the reduction in Type II error as the sample size increases. Notably, our Power method consistently achieves lower Type II error than Median across all sample sizes, particularly when the dimension of $Z$ is higher, underscoring its advantage. (For $d_Z = 4$ and 8 see Figures 5 and 6 in the Appendix)

**High-dimensional conditioning set $Z$.** We further conducted an experiment investigating the performance when the conditioning set $Z$ has an extremely high dimensionality. This is a task taken from (Polo et al., 2023), and the data is generated as follows:

$$Z \sim \mathcal{N}(0, d_Z), \quad Y = (Z^\top b)^2 + \mathcal{N}(0, 1), \quad X = Z^\top a + \gamma (Z^\top b)^2 + \mathcal{N}(0, 1) + cY,$$

where $a$ and $b$ were sampled from $\mathcal{N}(0, I_{d_Z})$, $c$ is a constant that determines the conditional dependence of $X$ and $Y$ on $Z$. We followed the setting of the hardest case in (Polo et al., 2023), choosing

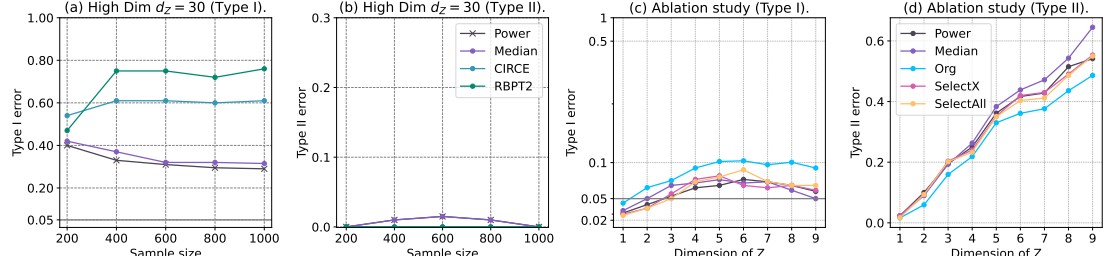

Figure 2: (a) and (b): Performance on high-dimensional conditional set data ($d_Z$ = 30). The $x$-axis represents the training sample size. (c) and (d): Ablation study on synthetic data with the significance level $\alpha = 0.05$. The $x$-axis represents the dimension of $Z$.

$d_Z$ = 30; H$_0$ : $\gamma = 0.02, c = 0$; H$_1$ : $\gamma = 0, c = 0.1$. In all cases, we used 200 test points with significance level $\alpha = 0.05$ and repeated the experiment 200 times for each training sample size.

Figure 2(a) and (b) present the results. One can see that all methods perform relatively poorly, failing to effectively control Type I error with Type II error nearly zero, indicating that all methods may not effectively block the influence of $Z$ on the dependent variable due to the estimated bias, thus incorrectly considering them conditionally independent. RBPT2 performs worse than kernel-based methods, as they almost reject all H$_0$. The kernel-based Median and Power performing better than CIRCE. And all methods show no convergence trend or have a very slow convergence rate, which may be due to the inherently low convergence rate of CME in challenging cases (Li et al., 2022). In such hard cases, the Type I error of Power is slightly lower than that of Median, both struggling to capture the true CI relationship.

### 4.1.2 ABLATION STUDY

We further analyzed the impact of different kernels involved in the statistic using our Power method, based on its variants: We first investigated the original form of KCI, which is the Hilbert-Schmidt norm of $\Sigma_{\ddot{X}Y|Z}$ (i.e. Eq. 2), denoted as *Org*. Next, we investigated the effectiveness of our proposed criteria in learning the kernel parameters applied to the regressed variables $X$ and $Y$. We adopted a two-step optimization process: (1) First, using the current kernel parameters $\boldsymbol{\theta}_t$ to obtain $K_{X|Z(t)}$ and $K_{Y|Z(t)}$ to estimate the conditional mean; (2) Then, calculate $\hat{J}_t$ (Eq. 8) with $K_{X|Z(t)}$ and $K_{Y|Z(t)}$, and update the parameters to obtain $\boldsymbol{\theta}_{t+1}$. This process is repeated iteratively. The parameters in $\boldsymbol{\theta}_t = [\sigma_x, \sigma_y, \sigma_z]$, where $\sigma_x$ represents the bandwidth involved in $\phi_x$ and the same applies to the rest, are initialized using the median heuristic and remain fixed during step 1. For each execution of step 1, step 2 is repeated 10 times, making a total of 10 iterations. The optimization algorithm and the corresponding learning rate remain at the default settings. Here, we adopted two variants: *SelectX*: only updating $\sigma_x$ and $\sigma_z$, and *SelectAll*: updating all the parameters in $\boldsymbol{\theta}_t$. These variants are compared with Median and Power on the synthetic data (Eq. 11) with sample size $n = 200$.

Figure 2(c) and (d) show the Type I and Type II errors of Power and its variants. *Org* represents the original form of KCI (Eq. 2) and involves the estimation of identity operator $\mu_{Z|Z}$, which is not Hilbert-Schmidt for characteristic RKHS, leading to significant estimating bias[2]. From the result, *Org* exhibits a higher rejection rate than the significance level of $\alpha = 0.05$, likely due to this estimation bias, compared to the decomposed Median. As a result, the influence of $Z$ on $X$ is not fully blocked by $\mu_{XZ|Z}$, causing the residuals to remain correlated with $Z$, which leads to a higher Type I error than expected and a tendency to reject H$_0$ due to the bias. Another observation is that the Type I and Type II errors of *SelectX* and *SelectXY* are mixed compared to Power, suggesting that updating $\sigma_x$ and $\sigma_y$ by maximizing the estimated power provides only marginal improvement over Power, which updates only $\sigma_z$. This could be attributed to the estimation bias of $\widehat{\mu}_{X|Z}$ and $\widehat{\mu}_{Y|Z}$ in practice. In the presence of these biases in the residuals, updating $\sigma_x$ and $\sigma_y$ does not result in the expected improvement in test power. Therefore, we choose to fix $\sigma_x$ and $\sigma_y$ using the median heuristic and

---

[2]In Mastouri et al. (2021, Appendix B.9 and Figure 4), for a 1D Gaussian $Z$, the CME estimator correctly captures the identity in high-density regions but becomes highly biased in the tail due to insufficient training data. A similar description is also provided in Li et al. (2022, Appendix D).

update only $\sigma_z$. This strategy enables our method to improve performance effectively with minimal additional computational cost, as we will discuss shortly.

### 4.1.3 TIME COMPLEXITY

Our Power differs from the original KCI primarily by decomposing $\phi_z$ and optimizing it according to step 4 in the overall test procedure. Consequently, our approach requires only one single estimation of the conditional means $\mu_{X|Z}$ and $\mu_{Y|Z}$ separately, similar to the original KCI, which remains the primary computational bottleneck. Therefore, the additional computational cost introduced by our method, mainly due to the learning of $\phi_z$, is minimal.

We conducted an experiment to analyze the computational cost of our method compared to Median. Table 1 presents the overall runtime of our method compared to Median for different sample sizes. The data was generated according to Eq. 11 with $Z$ having a dimension of 3. For each sample size, we randomly generated 20 cases. All experiments were conducted on the same device without GPU acceleration. From the results, it can be seen that optimizing the kernel parameter of $\phi_z$ almost does not incur additional computation cost compared to Median, as most of the time is spent on regression modeling and testing procedure. Therefore, overall, our Power method can automatically learn more suitable kernel parameter with minimal additional computational cost, thereby improving test accuracy. This allows our method to achieve improved performance over the original KCI in most scenarios where KCI (Zhang et al., 2011) is applied.

Table 1: Average testing time (s) ± standard deviation on different sample size.

| Sample Size | 100 | 200 | 500 | 1000 | 2000 |
|---|---|---|---|---|---|
| Power | 0.69±0.10 | 1.64±0.34 | 3.53±0.39 | 11.59±1.25 | 42.64±6.89 |
| Median | 0.46±0.04 | 1.58±0.14 | 3.43±0.36 | 11.31±1.54 | 41.66±7.88 |

### 4.2 REAL DATA

Following the setup of (Polo et al., 2023), we test our methods on the car insurance dataset originally collected from four US states and multiple insurance companies by Angwin et al. (2022), with three variables: car insurance price $X$, minority neighborhood indicator $Y$ and driver's risk $Z$. We follow the data assessment experiment from (Polo et al., 2023) to evaluate our method under a simulated $H_0$, where the driver risk $Z$ is divided into 20 bins and the $Y$ values corresponding to each bin are shuffled. With shuffled samples of $Z$, the evaluated method is expected to effectively control the average (over companies) rejection rate (i.e., Type I error) at the given significance level. The average error of CIRCE and RBPT2 is slightly below the given significance level. And Median and Power method are able to control the error rate relatively well. For more results and explanations regarding the experiment on car insurance dataset, please refer to Appendix B.3.

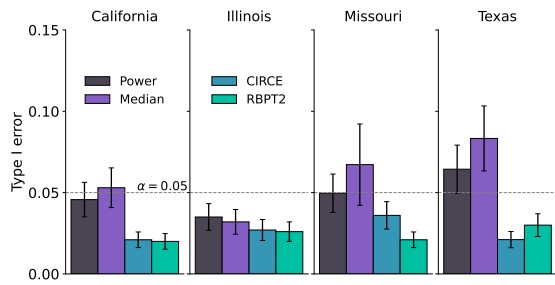

Figure 3: Type-I error result using *shuffled* car insurance data with significant level $\alpha = 0.05$ (dark line).

### 4.3 COMPARISON ON CAUSAL DISCOVERY

Our method can also be directly extended to causal discovery tasks (Glymour et al., 2019), improving upon the original KCI approach. Causal discovery aims to find causal structure from observational data, which is a fundamental scientific problem and has been extensively explored in various disciplines (see e.g. Zhang et al. (2018)). Formally, given $n$ random variables $X_1, X_2, \cdots, X_n$, causal discovery methods seek to depict the causal relationships among these variables through a directed

Table 2: Average F1 score ± standard deviation using PC as search algorithm on synthetic graph with different graph densities. Bold represents the better.

| Graph Density | 0.2 | 0.3 | 0.4 | 0.5 | 0.6 | 0.7 | 0.8 |
|---|---|---|---|---|---|---|---|
| Power | 0.781 ±0.061 | **0.685** ±0.088 | **0.648** ±0.086 | **0.580** ±0.065 | **0.503** ±0.059 | **0.409** ±0.078 | **0.441** ±0.049 |
| Median | **0.791** ±0.068 | 0.680 ±0.097 | 0.640 ±0.089 | 0.562 ±0.070 | 0.486 ±0.062 | 0.404 ±0.073 | 0.417 ±0.045 |

acyclic graph (DAG). CI testing serves as a core subroutine within constraint-based causal discovery methods (Pearl & Mackenzie, 2018). Constraint-based causal discovery methods like PC algorithm (Spirtes et al., 2000) make the additional assumption of faithfullness, wherein the joint distribution does not permit any CIs that are not entailed by the Markov condition[3]. So it is well known that small mistakes at the beginning of the algorithm (e.g. missing an independence relation) may lead to significant errors in the resulting DAG. Therefore the performance of those methods relies heavily on (conditional) independence tests. In this experiment, we compared the performance of Power with Median using the PC algorithm as the search method for causal discovery tasks.

We generated the synthetic causal graphs with varying graph densities ranging from 0.2 to 0.8. The graph density is measured by the ratio of the number of edges to the maximum possible number of edges in the graph; a smaller graph density indicates fewer edges in the graph, while a larger density indicates a denser graph. Each generated graph involves 10 variables with sample sizes of $n = 500$. For each variable $X_i$ in the graph, the data was generated according to $X_i = f_i(\text{PA}_i) + E$, where $\text{PA}_i$ are parent nodes of $X_i$ in the graph and $f_i$ were randomly chosen from the *linear*, *sin*, *cos*, *tanh*, *exponential* and *power* functions. For more implementation details, please refer to Appendix B.2.

We evaluate our Power and Median using F1 score[4]. A higher F1 score indicates higher accuracy. Table 2 shows the results. It can be observed that our Power outperforms Median in most graph density settings, except when the graph density is 0.2. This may be because the number of variables in the conditioning set increases along with the graph density. When the graph is relatively sparse with low graph density, the impact of kernel selection on $Z$ may not be evident with low-dimensional $Z$. As the dimension of conditioning set increases, our method can learn more suitable kernel parameters for these conditioning variables, leading to more accurate detection of CI relationships. Overall, our method can improve the performance of existing methods on the causal discovery task, particularly when the graph is dense.

## 5 CONCLUSION AND FUTURE WORK

In this paper, we propose a power-based kernel selection method to selectively learn the kernel parameters involved in KCI method for conditional independence test. These parameters are learned by maximizing the ratio of the estimated statistic to its variance, which is equivalent to maximizing test power in large sample sizes. We validate our method on synthetic data, real world benchmark, and causal discovery task. Experimental results demonstrate that our method outperforms the median heuristic-based approach on conditional independence tasks with minimal additional computational cost, suggesting that it can serve as a replacement for KCI in most CI-related task.

In the future, we aim to improve the regression process to reduce estimation bias. We aim to improve the regression with smaller bias to effectively learn the kernel parameters of the regressed variables using our proposed power-based criteria. This will enable us to further explore the optimal kernel choice for achieving a valid CI test with optimal power, bringing its performance in line with that observed in other kernel-based tasks.

---

[3]Markov condition assumes that the joint distribution satisfies all CIs that are imposed by the true causal graph. This is an assumption about the physical generating process of the data, not only about their distribution.

[4]F1 score is a weighted average of precision and recall, calculated as F1 $= \frac{2\text{recall}\cdot\text{precision}}{\text{recall}+\text{precision}}$.

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

APPENDICES

In this section, we provide further explanations about the testing procedure of our method in Appendix A.1 and the comparison baselines in Appendix A.2. We also present detailed experimental settings of our method B.1, synthetic dataset in Appendix B.2, real world dataset in Appendix B.3 and additional results in Appendix B.4.

## A  TESTING PROCEDURE AND BASELINES

### A.1  TESTING PROCEDURE

With $n$ observation points, the KCI statistic $C_{\mathrm{KCI}}^2$ has a biased HSIC-like estimator:

$$\widehat{C}_{\mathrm{KCI}_b}^2 = \frac{1}{n(n-1)} \mathrm{Tr}(H K_{X|Z} H (K_Z \odot K_{Y|Z})) \tag{12}$$

where $H = I - \frac{1}{n} 1_n 1_n^\top$ is the centering matrix. $\widehat{C}_{\mathrm{KCI}_b}^2$ has $O(1/n)$ bias and $O_p(1/\sqrt{n})$ deviation from the mean for any fixed probability of the deviation (see e.g. Pogodin et al. (2022), Lemma C.2). We first compute the residual covariance matrices $K_{X|Z}$ and $K_{Y|Z}$ and the kernel matrix $K_Z$ from $n$ testing data. We then denote $K = K_{X|Z}$ and $L = K_Z \odot K_{Y|Z}$. Then, we let the EVD decomposition of $K$ and $L$ be $K = V_K \Lambda_K V_K$ and $L = V_L \Lambda_L V_L$. $\Lambda_K$ (resp. $\Lambda_L$) is the diagonal matrix containing non-negative eigenvalues $\lambda_{K,i}$ (resp. $\lambda_{L,i}$). Let $\boldsymbol{\psi}_K = [\psi_{K,1}(\boldsymbol{x}), \cdots, \psi_{L,n}(\boldsymbol{x})] = V_K \Lambda_K^{1/2}$ and $\boldsymbol{\phi}_L = [\phi_{L,1}(\boldsymbol{y}, \boldsymbol{z}), \cdots, \phi_{L,n}(\boldsymbol{y}, \boldsymbol{z})] = V_L \Lambda_L^{1/2}$. And its null distribution can be approximated in two ways: as (1) weighted (infinite) sum of $\chi^2$ variables, or through (2) Gamma approximation.

**Weighted sum of $\chi^2$ approximation.** Under $H_0$, $X \perp\!\!\!\perp Y \mid Z$, $\widehat{C}_{\mathrm{KCI}_b}^2$ has the same asymptotic distribution as

$$\check{T}_b = \frac{1}{n(n-1)} \mathrm{Tr} \sum_{k=1}^{n^2} \tilde{\lambda}_k \cdot z_k^2, \tag{13}$$

where $z_k \sim \mathcal{N}(0,1)$ and where $\tilde{\lambda}_k$ are eigenvalues of $\mathbf{w}\mathbf{w}^\top$ and $\mathbf{w} = [\mathbf{w}_1, \cdots, \mathbf{w}_n]$, with the vector $\mathbf{w}_t$ obtained by stacking $\boldsymbol{M}_t = [\psi_{K,1}(x_t), \cdots, \psi_{K,n}(x_t)]^\top \cdot [\phi_{L,1}(y_t, z_t), \cdots, \phi_{L,n}(y_t, z_t)]$. This conclusion primarily relies on the continuous mapping theorem, for details refer to (Zhang et al., 2011, Theorem 3).

**Gamma approximation.** Following (Gretton et al., 2007), the null distribution for the KCI estimator $\frac{1}{n^2} \mathrm{Tr}(KL)$ can also be approximated by a Gamma distribution, which is $p(t) = t^{k-1} \frac{e^{-t/\theta}}{\theta^k \gamma(k)}$, with the parameters

$$k = \frac{\mu}{\sigma^2}, \quad \theta = \frac{\sigma^2}{n\mu}, \quad \text{with} \quad \mu = \frac{1}{n} \mathrm{Tr}(\mathbf{w}\mathbf{w}^\top) \quad \text{and} \quad \sigma^2 = 2 \frac{1}{n^2} \mathrm{Tr}[(\mathbf{w}\mathbf{w}^\top)^2]. \tag{14}$$

Therefore, one can use Monte Carlo simulation to approximate the null distribution according to the two approaches mentioned above. The complete testing procedure is as follows: we first estimate the conditional means $\mu_{X|Z}$ and $\mu_{Y|Z}$ and learn the parameters in $\phi_z$ on the training data and calculate the $K_{X|Z}$, $K_{Y|Z}$ and $K_Z$ on the testing data and the eigenvalues and eigenvectors of $K$ and $L$ defined above. Then we evaluate the statistic $\widehat{C}_{\mathrm{KCI}_b}^2$ according to Equation 12. And then we simulate the null distribution either by (1) weighted sum of $\chi^2$ approximation (according to Eq. 13) or (2) Gamma approximation (with the parameters given by Eq. 14). We then obtain a set of statistics $\check{\boldsymbol{T}} = (\check{T}_1, \cdots, \check{T}_m)$ through sampling. Then the $p$-value is calculated as the proportion of the statistic $\check{T}_j$ in $\check{\boldsymbol{T}}$ that is greater than $\widehat{C}_{\mathrm{KCI}_b}^2$. Finally, if the $p$-value is not greater than the given significance level $\alpha$, we reject $H_0$ and hold $H_1$; otherwise, we hold $H_0$.

### A.2  CONDITIONAL INDEPENDENCE TESTING BASELINES

**CIRCE** (Conditional Independence Regression CovariancE, (Pogodin et al., 2022)) is a simplified version of KCI, which only considers the correlation between $\phi_y(Y)$ and the regression residuals of

$Z$ to $\phi_{xz}(X, Z)$, i.e. $\phi_{\ddot{x}|z}(X, Z) = \phi_{\ddot{x}}(X, Z) - \mathbb{E}[\phi_{\ddot{x}}(X, Z) \mid Z]$ with $\ddot{X} = (X, Z)$. As explained in Theorem 2, any function $g(X, Z) \in L^2$ can capture the general relationship between $X$ and $Z$. Utilizing the reproducing property, the residual feature map $\phi_{\ddot{x}|z}(X, Z)$ effectively eliminates the influence of $Z$ on $X$. Intuitively, this residual $\phi_{\ddot{x}|z}(X, Z)$ thus represents the component of $X$ that cannot be explained by $Z$. Thus, if $\phi_{\ddot{x}|z}(X, Z)$ is independent of $Y$, then we can conclude that $X$ and $Y$ are conditionally independent given $Z$. Formally, CIRCE has the following form:

$$T_{\text{CIRCE}} = \mathbb{E}[\phi_z(Z) \otimes \phi_y(Y) \otimes (\phi_x(X) - \mu_{X|Z}(Z))]. \tag{15}$$

Correspondingly, CIRCE also has an MMD-like biased estimator:

$$\widehat{T}_{\text{CIRCE}} = \frac{1}{n(n-1)} \text{Tr}(H K_Z H (K_Y \odot K_{X|Z})). \tag{16}$$

and can similarly use weighted (infinite) sum of $\chi^2$ variables or Gamma approximation to estimate the null distribution for conducting CI testing.

In CIRCE, we follow the original settings of CIRCE: we use the median heuristic to initialize the parameters of $\phi_z$, $\phi_y$ and $\phi_x$. We also use a Gaussian process to estimate the conditional mean embedding $\mu_{X|Z}$, with parameters set identical to those used in our Power method.

**RBPT2** (The Rao-Blackwellized Predictor Test, (Polo et al., 2023)) involve a regression chain: it first needs to estimate $g(Y, Z) = [X \mid Y, Z]$. Then with the trained $g(Y, Z)$, it estimates $h(Z) = [g(Y, Z) \mid Z]$. The statistic is defined to compare the difference between their predicted results and the residuals of the real value of $X$, which is

$$T_i = l(h(z_i), x_i) - l(g(y_i, z_i), x_i), \quad S = \frac{\sqrt{n} \sum_{i=1}^n T_i}{\sqrt{\left(\frac{1}{n} \sum_{i=1}^n (T_i)^2 - \left(\frac{1}{n} \sum_{i=1}^n T_i\right)^2\right)}},$$

where $l$ is MSE loss $l = (g - x)^2$ and its $p$-value The p-value is then computed as $p = 1 - \Phi(S)$. We follow its default model and parameter setting[5].

# B EXPERIMENTAL DETAILS

We present the implementation details of both our proposed method and the synthetic dataset for conditional independence test and causal discovery tasks.

## B.1 IMPLEMENTATION DETAILS

Our method's parameters mainly exist in the kernel ridge regression, the process of learning the parameters in $\phi_z$ and the final testing procedure. (1) For the kernel ridge regression, there are three parameters trainable, the amplitude $A$, the bandwidth involved in $K_Z^R$, denoted as $\sigma_z^r$, and the regularization parameter $\varepsilon$. To ensure stability during the training process, we have constrained their value ranges, the amplitude $A$ is limited to the range of $[10^{-3}, 10^3]$. The bandwidth $\sigma_z^r$ is a vector whose dimensions are the same as those of conditioning variable $Z$, with values constrained to $[10^{-2}, 10^2]$. The regularization parameter $\varepsilon$ is constrained to $[10^{-10}, 1]$. We use marginal likelihood as the loss function and the L-BFGS-B algorithm (Liu & Nocedal, 1989) to optimize and update these parameters.

(2) For the kernel parameters on $\phi_z$, the bandwidth $\sigma_z$ is also a vector of the same dimension as $Z$. We apply $\lambda$ to the estimated variance to avoid numerical issues, i.e., $\widehat{\sigma}_1 = \sqrt{\widehat{\sigma}_1^2 + \lambda}$ with $\lambda = 10^{-10}$. We adopt Adam (Kingma & Ba, 2014) as the optimization algorithm for this parameter over 100 iterations with learning rate $lr = 0.01$. (3) In the final test stage, we use the weighted sum of $\chi^2$ approximation to simulate the null distribution. Following the default setting in (Zhang et al., 2011), we drop all $\tilde{\lambda}_k$ which are smaller than $10^{-5}$ for computational efficiency. We sampled a total of 1000 $\check{T}_b$ values according to Eq. 13, and obtained the $p$-value which is the rate that $\check{T}_b > \widehat{C}_{\text{KCI}_b}^2$.

---

[5] https://github.com/felipemaiapolo/cit

## B.2 MORE DETAILS ON SYNTHETIC DATA

**Implementation details of Synthetic CI dataset.** In the CI testing task, we assume $X$ and $Y$ are the dependent variable of $Z$. To examine Type I errors, $X$ and $Y$ were generated according to the following post-nonlinear function model:

$$X = g(\sum_i f_i(Z_i) + E), \tag{17}$$

where $f_i$ and $g$ were randomly chosen from the *linear*, *sin*, *cos*, *tanh* and power function. The *linear* function has three options: $1.25, 1.7$ and $2.5$. For the power function, the power $a$ in $x^a$ is randomly selected from $1, 2, 3$. For each function class in $f_i$ and $g$, they all have the same probability of being selected, and within each corresponding class, the parameter set has an equal probability of being selected (e.g., the probability of selecting a *linear* function with a weight of $1.25$ is $\frac{1}{5} \times \frac{1}{3}$). The noise term $E$ was randomly generated from either a normal distribution $\mathcal{N}(0, 0.5)$ or a uniform distribution $U(-0.5, 0.5)$ with equal probability. The conditioning variable $Z$ is generated following $Z \sim \mathcal{N}(0, 1)$. To test Type II error, we add the same latent variable $T$ to both $X$ and $Y$ with $T \sim \mathcal{N}(0, 0.5)$. Then the dependent variable, e.g. X, is generated as follows:

$$X = g(\sum_i f_i(Z_i) + E) + T, \tag{18}$$

$Y$ follows the same generating process with the same variable $T$. In the experiment using this synthetic dataset, the amount of testing data is the same as the training data.

**Implementation details of graph dataset for causal discovery.** In the synthetic graph data for causal discovery task, each generated graph involves 10 variables with sample sizes of $n = 500$, which are evenly divided into training data and testing data. For each variable $X_i$ in the graph, the data was generated according to

$$X_i = f_i(PA_i) + E,$$

where $PA_i$ represents the parent nodes of $X_i$ in the graph. $f_i$ is equally likely to be sampled from *linear, sin, cos, tanh, exp* and $x^\alpha$. The *linear* function has two weight options: $0.5$ and $2.5$, and $\alpha$ in $x^\alpha$ is randomly selected from $1, 2, 3$. Each function class in $f_i$ all has the same probability of being selected, and within the equal probability of each parameters setting. If one of the variables has no parent nodes in the graph, it follows a standard normal distribution. $E$ represents the noise variable, randomly following either a Gaussian distribution with $\mathcal{N}(0, 0.5)$ or a uniform distribution $U(-0.5, 0.5)$ with equal probability. For each graph density, we generated 20 realizations. We set the significance level of $\alpha = 0.10$.

## B.3 REAL DATA

The car insurance data[6] encompasses four US states (California, Illinois, Missouri and Texas) and includes information from numerous insurance providers compiled at the ZIP code granularity. The data offers a risk metric and the insurance price levied on a hypothetical customer with consistent attributes from every ZIP code. ZIP codes are categorized as either minority or non-minority, contingent on the percentage of non-white residents. The variables in consideration are $Z$, denoting the driving risk; $X$, an indicator for minority ZIP codes; and the insurance price $Y$. A pertinent question revolves around the validity of the null hypothesis $H_0 : X \perp\!\!\!\perp Y \mid Z$, essentially questioning if demographic biases influence pricing.

Since this is a real dataset, the full distribution and the true CI relationship between $X$ and $Y$ given $Z$ are unknown. Therefore, following (Polo et al., 2023), we discretize the conditioning variable $Z$ into twenty distinct values and shuffle the $Y$ values corresponding to each discrete $Z$ value. If a test maintains Type-I error control, we expect it to reject $H_0$ for at most $\alpha = 0.05$ of the companies in each state. In the second part, we use the unshuffled data for CI testing and focus on assessing the power of our methods. Following the default setting in Polo et al. (2023), the dataset is split $70/30\%$ for training and testing. We conducted a total of 5 experiments, each time randomly selecting 10 seeds, and reported the average Type-I error rate and the average $p$-value. We evaluated the performance of our method and the comparison methods, Median, CIRCE (Pogodin et al., 2022), and RBPT2

---

[6]Data description link

(Polo et al., 2023), in this simulation experiment, with the parameters being the same as those set in synthetic data experiment.

Figure 7 shows that all methods control Type-I errors relatively well: Power and Median exhibit slightly higher Type-I errors in Missouri and Texas, while CIRCE remains slightly below the significance level across all four states. Compared to Median, Power provides slightly better control. Figure 8 presents the test results on the original unshuffled data. All methods show relatively low p-values, leading to the conclusion that all states likely exhibit varying degrees of discrimination against minorities in ZIP codes. The severity, in descending order, is Illinois, Texas, Missouri, and California. This result is consistent with the findings from (Angwin et al., 2022), indicating that our method is capable of correctly identifying CI relationships in the real world.

## B.4 MORE EXPERIMENTAL RESULTS

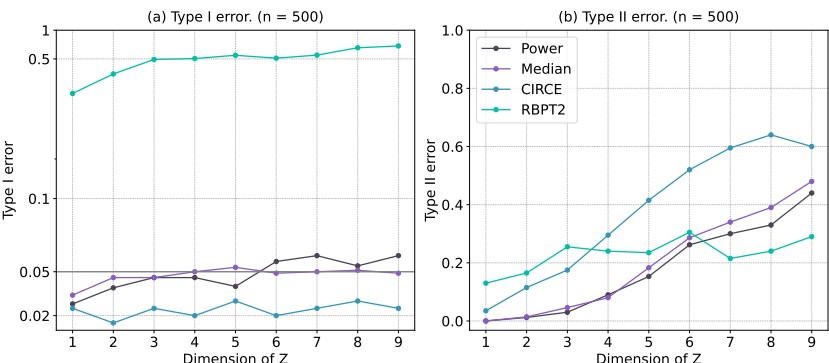

Figure 4: Type I error (a) and Type II error (b) on synthetic data with the significance level $\alpha = 0.05$ (gray line) when increasing the dimension of conditioning variable $Z$, keeping sample size $n = 500$.

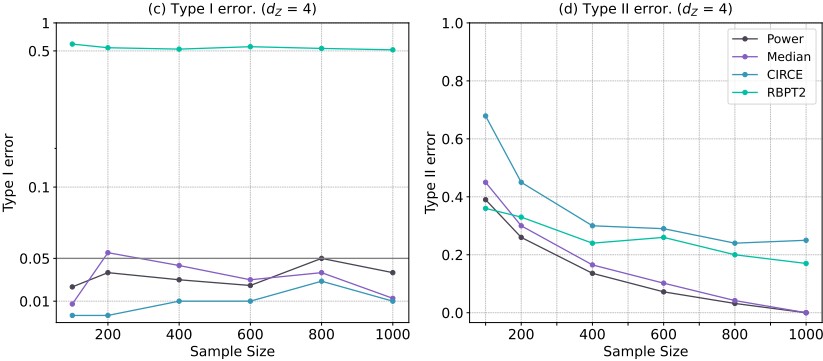

Figure 5: Type I error (a) and Type II error (b) on synthetic data with the significance level $\alpha = 0.05$ (gray line) when increasing the number of samples, keeping the dimension $d_Z = 4$.

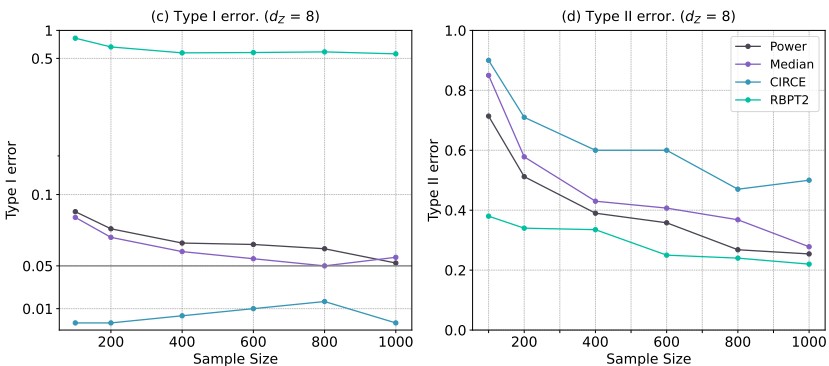

Figure 6: Type I error (a) and Type II error (b) on synthetic data with the significance level $\alpha = 0.05$ (gray line) when increasing the number of samples, keeping the dimension $d_Z = 8$.

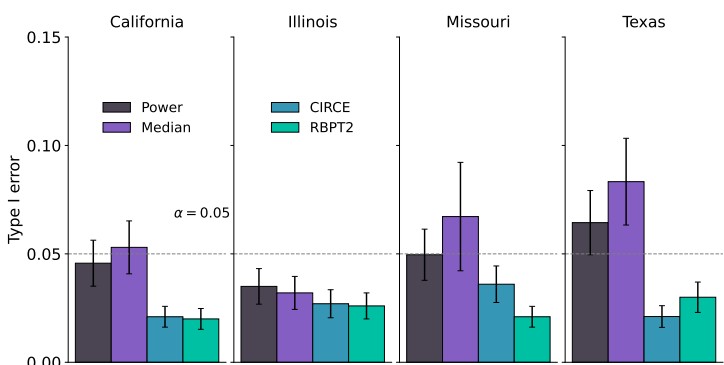

Figure 7: Performance on *shuffled* car insurance dataset.

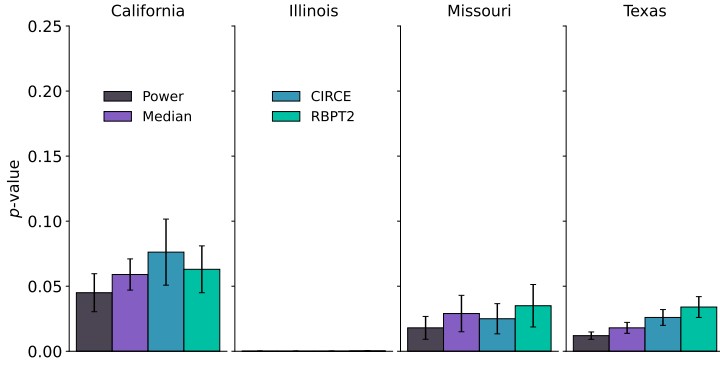

Figure 8: Performance on *unshuffled* car insurance dataset.

