# OpenReview forum: "Practical Kernel Learning for Kernel-based Conditional Independent Test"
_ICLR.cc/2025/Conference — Submitted to ICLR 2025_

### Official Review · Reviewer_QG5Q · 2024-10-27

**Soundness:** 2
**Presentation:** 3
**Contribution:** 2
**Rating:** 5
**Confidence:** 4

**Summary:**

The paper presents a method, termed POWER, designed for optimizing kernel parameters to improve the performance of kernel-based conditional independence tests. This method enables users to learn kernel parameters directly from the data, thereby enhancing the test's power. Experimental results demonstrate that it also positively impacts constraint-based causal discovery methods.

**Strengths:**

1. In contrast to previous methods that generally employ a median heuristic algorithm for selecting kernel parameters, this paper introduces a data-driven approach to identify the optimal parameters, thereby enhancing the test's power to some extent.

2. This study examines the effect of optimizing kernel parameters for various variables on the algorithm's power. Ultimately, the optimization is confined to the kernel function within the conditioning set, which maximizes the algorithm's power while maintaining manageable time complexity.

3. The experimental results demonstrate that the proposed POWER method outperforms both the original kernel parameter settings and several conditional independence baselines in certain settings. Furthermore, when applied to causal discovery, it generally produces superior results.

**Weaknesses:**

1. The method presented in this paper demonstrates only marginal improvements in power, yielding minimal gains compared to the original median heuristic algorithm.

2. The conditional mean of the statistic is estimated using kernel ridge regression; however, the paper derives the asymptotic normality of the oracle statistic only under the case when the conditional mean is known.

3. The proposed method has not been evaluated in scenarios characterized by higher dimensionality and smaller sample sizes. The maximum tested dimensionality of Z is only 30, and the minimum sample size is 200, which restricts the significance of the results.

**Questions:**

1. If the estimated conditional mean is incorporated into the statistic, does it still retain asymptotic normality? What conditions are necessary for it to maintain this property? If asymptotic normality is preserved, how does it affect the asymptotic variance? If the asymptotic variance changes, the methods for enhancing power derived from the original asymptotic variance may no longer be valid.

2. In the context of estimating conditional means for conditional independence testing, several more effective methods have emerged. Notably, generative models can serve this purpose. For instance, Shi et.al. (2021) employed Sinkhorn GAN for estimating conditional means. If these methods were to replace kernel ridge regression for estimating conditional means, what impact would this have on your approach?

3. Your method seems to overlook scenarios involving higher dimensionality of Z and smaller sample sizes. For example, dz=100 and n=500 as in Li et.al. (2024).

4. In addressing the causal discovery problem, you have only presented the F1 scores for the POWER and Median methods, and the results appear to lack significance. Could you conduct additional experiments to provide further clarification?

5. Could you provide a theoretical explanation of how this method enhances statistical power? For example, you can derive its asymptotic power against local alternative under a specific model (Niu et.al. (2022)).

6. In theoretical frameworks, it is often essential to split samples—utilizing one subset to estimate the conditional mean and another to compute the statistic (Shi et.al. (2021), Shah et.al. (2020)). However, in practice, sample splitting may result in a reduction in statistical power. Did the authors evaluate the effects of data splitting on their methodology through simulation experiments to determine whether it is necessary to perform data splitting?

Reference:

1.	Shi, C., Xu, T., Bergsma, W., & Li, L. (2021). Double generative adversarial networks for conditional independence testing. Journal of Machine Learning Research, 22(285), 1-32.

2.	Niu, Z., Chakraborty, A., Dukes, O., & Katsevich, E. (2022). Reconciling model-X and doubly robust approaches to conditional independence testing. arXiv preprint arXiv:2211.14698.

3.	Shah, R. D., & Peters, J. (2020). The hardness of conditional independence testing and the generalised covariance measure.

4.	Li, S., Zhang, Y., Zhu, H., Wang, C., Shu, H., Chen, Z., ... & Yang, Y. (2024). K-nearest-neighbor local sampling based conditional independence testing. Advances in Neural Information Processing Systems, 36.

---

> ### Author Response · Authors · 2024-11-23
>
> We sincerely appreciate your profound and valuable suggestions for our paper. Thank you for your effort and dedication!
>
> >**W1.** The method presented in this paper demonstrates only marginal improvements in power, yielding minimal gains compared to the original median heuristic algorithm.
>
> **A.** We believe that evaluating our method requires considering its minimal additional computational overhead. Experimental results show that our method consistently outperforms the Median heuristic-based approach across different dimensions of the conditioning set. This also demonstrates that our method can effectively select more suitable kernel parameters.
>
> >**W2. & Q1.** The conditional mean of the statistic is estimated using kernel ridge regression; however, the paper derives the asymptotic normality of the oracle statistic only under the case when the conditional mean is known. If the estimated conditional mean is incorporated into the statistic, does it still retain asymptotic normality?
>
> **A.** Our current method uses Gaussian processes to estimate the conditional mean. While it does have estimation bias, the estimation error decreases as the sample size increases (even though its learning rate is complex), which improves the accuracy of the test power. Therefore, in cases with larger sample sizes, we can still utilize the approximate testing power derived from this property for parameter learning. From a theoretical perspective, due to the unclear estimation error of the CME, further research is required. This will be our future research direction. Thank you very much for your suggestion.
>
> >**W3. & Q3.** Your method seems to overlook scenarios involving higher dimensionality of Z and smaller sample sizes. For example, dz=100 and n=500 as in Li et.al. (2024).
>
> **A.** Thank you for your suggestion. We followed the experimental setting in Li et al. (2024) to test performance under different sample sizes and compared it with the baseline proposed by Li et al. The results are as follows.  Except for Li's method, which retains all default settings, the data in rest methods was evenly split into training and test datasets. From the results, it can be observed that, even in high-dimensional scenarios, under certain specific settings, the kernel-based methods still demonstrate comparable Type I error control and test power.
>
> Type I Error with different sample size N ($\alpha = 0.05$)
>
> | **N** | CIRCE | Median| Power | KNN |
> |------------------|-----------|------------|-----------|---------|
> | 100              | 0         | 0.02       | 0.02      | 0.05    |
> | 300              | 0.08      | 0.04       | 0.04      | 0.04    |
> | 500              | 0.11      | 0.04       | 0.04      | 0.05    |
>
> Type II Error
>
> | **N** | **CIRCE** | **Median** | **Power** | **KNN** |
> |------------------|-----------|------------|-----------|---------|
> | 100              | 0.14      | 0.47       | 0.46      | 0.69    |
> | 300              | 0         | 0.03       | 0.01      | 0.26    |
> | 500              | 0         | 0          | 0         | 0.09    |
>
>
> >**Q2.** Generative models can serve this purpose such as Sinkhorn GAN for estimating conditional means. If these methods were to replace kernel ridge regression for estimating conditional means, what impact would this have on your approach?
>
> **A.** Thank you for your suggestion. Our method regularizes CME in the RKHS space instead of in the real number space. Using GANs to estimate the theoretically infinite-dimensional CME may require further consideration.

---

> ### Author Response · Authors · 2024-11-23
>
> >**Q4.** In addressing the causal discovery problem, you have only presented the F1 scores for the POWER and Median methods, and the results appear to lack significance. Could you conduct additional experiments to provide further clarification?
>
> **A.** Thank you for your suggestion. We further evaluated our method and the Median approach on two real-world benchmarks (SACHs and CHILD), using the same experimental parameters as those in our synthetic dataset experiments. The results are as follows. From the results, it can be observed that on these two graphs, our method also outperforms the Median-based approach.
>
> Table 1: SACHS Results
>
> | **Metric** | F1 (N = 200)      | SHD (N = 200)     | F1 (N = 500)      | SHD (N = 500)     |
> |------------|------------------------|-----------------------|-----------------------|-----------------------|
> | Median     | $\underline{0.340} \pm 0.04$      | $\underline{17.83} \pm 0.66$      | $0.491 \pm 0.04$      | $17.19 \pm 0.60$      |
> | Power      | $0.328 \pm 0.06$      | $18.04 \pm 0.55$      | $\underline{0.519} \pm 0.03$      | $\underline{16.54} \pm 0.82$      |
>
> Table 2: CHILD Results
>
> | **Metric** | F1 (N = 200)      | SHD (N = 200)     | F1 (N = 500)      | SHD (N = 500)     |
> |------------|------------------------|-----------------------|-----------------------|-----------------------|
> | Median     | $0.492 \pm 0.02$      | $18.24 \pm 0.45$      | $0.762 \pm 0.03$      | $19.38 \pm 0.52$      |
> | Power      | $\underline{0.543} \pm 0.03$      | $\underline{17.67} \pm 1.28$      | $\underline{0.780} \pm 0.03$      | $\underline{18.76} \pm 0.71$      |
>
>
> >**Q5.** Could you provide a theoretical explanation of how this method enhances statistical power?
>
> **A.** Thank you for your comments. The theoretical analysis of test power involves the estimation bias of CME in our method, which requires further investigation. We appreciate you providing us with a good alternative approach for analyse, and we will attempt further theoretical analysis in this direction.
>
> >**Q6.** Did the authors evaluate the effects of data splitting on their methodology through simulation experiments to determine whether it is necessary to perform data splitting?
>
> **A.** Our method indeed requires splitting the samples, as it involves not only estimating the conditional mean but also learning the parameters of the statistic itself (specifically, the kernel parameters applied to the conditioning set). Below is an experiment demonstrating the impact of data splitting on our method. It shows a significant increase in Type I error when the data is not split, indicating that the learned parameters are overfitted to the training sample size. Therefore, splitting the data is essential for the validity of our method. In the experiment, the training set and test set were evenly split.
>
> Table: Comparison of Type I and Type II Errors (Split and Non-Split)
>
> | **Sample size**         | **100** | **200** | **400** | **600** |
> |-------------------------|-------------|-------------|-------------|-------------|
> | Type I (Split)         | 0.04        | 0.05        | 0.06        | 0.05        |
> | Type I (Non-Split)     | 0.01        | 0.10        | 0.13        | 0.11        |
> | Type II (Split)        | 0.56        | 0.36        | 0.26        | 0.17        |
> | Type II (Non-Split)    | 0.32        | 0.18        | 0.11        | 0.07        |

---

> > ### Comment · Reviewer_QG5Q · 2024-11-26
> > **Comments**
> >
> > W1
> > Although your method only slightly increases computation time, the improvement compared to the median is still not significant. Especially when d_z<5, there is almost no improvement.
> >
> > W2 & Q1
> > In the paper's "overall test procedure," the authors wrote: Using kernel ridge regression (Bach & Jordan, 2002) to estimate μ_(X|Z) and μ_(Y|Z). Why is it then mentioned that Gaussian processes are used to estimate the conditional mean? Are Gaussian processes only used to train parameters in the model? Could the authors provide a sufficient condition for the convergence rate of CME estimation similar to Theorem 6 in Shah, R. D., & Peters, J. (2020) and then derive its asymptotic distribution?
> >
> > W3 & Q3
> > Could the authors indicate which model from Li et. al. (2024) was adopted? Is it the post-nonlinear model? Is the noise Gaussian or Laplace? For different values of N, is the dimension of Z always 100? Under this setup, your proposed method does not appear to show a significant improvement compared to the median.
> >
> > Q4
> > Thank you for providing the additional results. However, in the SACHs results, when N=200, the median's F1 and SHD are slightly higher than those of Power. It is hard to say that your proposed method outperforms the median-based approach in this case.
> >
> > Q6
> > Thank you for providing the additional results; it makes sense to me. However, why is the Type I error lower for the non-split results compared to the split-sample results when the sample size is 100? Could the authors explain the reason?

---

### Official Review · Reviewer_BJCn · 2024-11-03

**Soundness:** 3
**Presentation:** 3
**Contribution:** 2
**Rating:** 5
**Confidence:** 3

**Summary:**

This paper addresses the fundamental and challenging problem of conditional independence testing in statistics and machine learning. The authors propose to optimize kernel parameters by maximizing the ratio of the estimated Kernel-based Conditional Independence (KCI) test statistic to its variance. They assert that their approach enhances test power with minimal additional computational cost. Through extensive experiments, they demonstrate the effectiveness of this approach in accurately performing conditional independence tests.

**Strengths:**

The proposed method can learn the kernel parameters with increased test power at a small additional computation cost.

**Weaknesses:**

1. I think the main weakness of this paper lies in its novelty. The general idea is quite similar to Gretton et al. (2012). Though the authors claim that the CI testing requires indirectly considering the correlations between residuals rather than the original data, I don't see any specific methodological designs that incorporate this when optimizing kernel parameters.

2. The proposed method does not appear to show significant improvement based on the results presented in Figures 1 and 2.

**Questions:**

1. The authors suggest to optimize kernel parameters by maximizing the ratio of the estimated Kernel-based Conditional Independence test statistic to its variance. What is the convergence rate of these estimated parameters? Do they affect the asymptotic variance in equation (6)? Are there any theoretical guarantees of the proposed method?

2. In Section 2.1, the authors mention the necessity of a balanced assessment of both Type I and Type II error rates in the practical evaluation of CI methods. While I acknowledge the difficulty of CI testing problems, as highlighted by Shah and Peters (2020), I argue that controlling Type I error rates is paramount. Under what conditions can Type I error be effectively controlled by the proposed method?

---

> ### Author Response · Authors · 2024-11-23
>
> We sincerely appreciate your profound and valuable suggestions for our paper. Thank you for your effort and dedication!
>
> >**W1.**  I think the main weakness of this paper lies in its novelty. I don't see any specific methodological designs that incorporate this when optimizing kernel parameters.
>
> **A.** Thank you for your comment. It is true that using test power to optimize kernel parameters has been widely adopted since Gretton et al. (2012) proposed it. However, to the best of our knowledge, the choice of kernels in CI tasks has not been thoroughly studied. Generally, this methodology optimizes all kernel parameters involved in the statistic. However, due to the unique nature of CI tasks (where the estimated conditional mean embedding inherently contains bias with limited samples), we found that directly optimizing the original KCI statistic does not effectively improve its test power.
>
> Instead, we took a surrogate approach by decomposing the original KCI statistic and directly exposing the kernel parameters applied to the conditional variable for optimization. This partially improved the test power. While we cannot claim to have found the optimal kernel parameters as Gretton et al. (2012) did, this approach avoids the redundancy of re-learning the conditional mean embedding by selecting the kernel parameters on the response variable. In terms of results, our method achieves higher test power with minimal computational overhead and comparable Type I error control, which represents our improvement.
>
> >**W2.** The proposed method does not appear to show significant improvement based on the results presented in Figures 1 and 2.
>
> **A.** We aim to propose a kernel parameter selection method for KCI that outperforms the median heuristic. As shown in Figure 1, our method consistently performs better than the median heuristic across different dimensions and sample sizes with comparable Type I control. Furthermore, considering the minimal additional computational overhead of our approach, we believe that our method can replace the original median heuristic-based KCI in most cases.
>
> >**Q1.** What is the convergence rate of these estimated parameters? Do they affect the asymptotic variance in equation (6)? Are there any theoretical guarantees of the proposed method?
>
> **A.** Due to the presence of conditional embedding bias, whose convergence rate remains a challenge, the convergence rate of the estimated statistic and the parameters involved is actually influenced by the convergence rate of the conditional expectation and, therefore, is still under further investigation.
> However, as shown in the experimental results, under a reasonably large sample size, our method can use the estimated test power based on the asymptotic distribution to select more suitable kernel parameters using the estimated power.
> We will focus more on the convergence rate of these parameters in future work. Thank you for your comments.
>
> >**Q2.** In Section 2.1, the authors mention the necessity of a balanced assessment of both Type I and Type II error rates in the practical evaluation of CI methods. While I acknowledge the difficulty of CI testing problems, as highlighted by Shah and Peters (2020), I argue that controlling Type I error rates is paramount. Under what conditions can Type I error be effectively controlled by the proposed method?
>
> **A.** Thank you for your comments. "The necessity of a balanced assessment of both Type I and Type II error rates in the practical evaluation of CI methods" are grounded in a "practical" perspective. On one hand, CI testing inherently differs from other hypothesis testing tasks, such as two-sample tests or independence tests, because it specifically considers whether there is a shared latent confounder among observed variables. This involves testing the independence of latent variables, requiring the exposure of latent variables (which introduces misspecification bias due to limited samples in practice). Thus, without assumptions, no method can effectively control the Type I error while maintain valid test power when dealing with general CI cases in practice. On the other hand, while some approaches like GCM (Shah et al., 2020) and CRT (Candes et al., 2018) claim to effectively or asymptotically control Type I error, they often come with stringent conditions (e.g., assuming the conditional distribution is known or the in-sample mean-squared error is small), which is almost impossible to achieve or verify in practice.
>
>
> In our method, due to the estimation of CME involved, which affects the control of Type I error, its convergence behavior is still unclear and requires further investigation. But from a practical perspective, our method empirically allows for better selection of kernel parameters across general cases. This demonstrates that even in the presence of CME estimation bias, our method remains effective and achieves better test power compared to the Median approach from a practical perspective.

---

### Official Review · Reviewer_U6fH · 2024-11-04

**Soundness:** 3
**Presentation:** 3
**Contribution:** 2
**Rating:** 3
**Confidence:** 4

**Summary:**

This paper focuses on the parameter selection problem for kernel-based methods in conditional independence testing. By applying the decomposed statistic of kernel-based conditional independence test, the authors learn the kernel parameters by maximizing the ratio of the estimated statistic to its variance, which approximates the test power at large sample sizes.

**Strengths:**

This paper proposes a kernel parameter selection approach for the kernel-based conditional independence test. Appropriate experiments are conducted with some comparative methods.

**Weaknesses:**

The idea of learning kernel functions or kernel parameters is not new in the literature. Related papers include "Optimal kernel choice for large-scale two-sample tests", "Learning Deep Kernels for Non-Parametric Two-Sample Tests", and "Generative models and model criticism via optimized maximum mean discrepancy". This diminishes the novelty of the current paper.

**Questions:**

1. From my understainding, the feature map should be denoted by $\phi(x): \mathcal{X} \to \mathcal{H}_{\mathcal{X}}$.

2. Section 3 includes several components, but it is unclear which parts are original contributions by the authors and which are drawn from existing literature. Please clarify this distinction, and provide appropriate references for any cited results.

3. Does the proposed method learn only the kernel bandwidths, or does it also learn the kernel functions? A more thorough discussion of the novelty of this method compared to existing approaches is necessary.

4. The asymptotic results are derived under fixed kernel parameters. How do the learned parameters affect these results? Moreover, how can the authors ensure that the proposed method, which uses the weighted sum of chi-squared to compute the $p$-value, controls the type I error at least asymptotically?

5. Could the authors provide a more detailed explanation of the impact of the conditional expectation bias?

6. Some phrases lack precision. For example, in "and Type II errro generally rises with increasing $Z$" and "slightly exceeding the significance level as $Z$ increases", it should refer to the increasing dimensionality of $Z$. In "$Z \sim N(0, d_Z)$", does $d_Z$ represent the dimension of $Z$?

7. Does the original form of KCI in Section 4.1.2 use the median heuristic for determining kernel bandwidth?

8. Does the discreteness of continuity of $X, Y, Z$ affect the performance of the proposed method, such as the choice of kernel?

---

> ### Author Response · Authors · 2024-11-23
>
> We sincerely appreciate your profound and valuable suggestions for our paper. Thank you for your effort and dedication!
>
> >**W.** The idea of learning kernel functions or kernel parameters is not new in the literature.
>
> **A.** Thank you for your comment. Using test power to optimize kernel parameters has been widely adopted since Gretton et al. (2012) proposed it. However, to the best of our knowledge, the choice of kernels in CI tasks has not been thoroughly studied.
> Generally, this methodology optimizes all kernel parameters involved in the statistic. However, due to the unique nature of CI tasks (where the estimated conditional mean embedding inherently contains bias with limited samples), we found that directly optimizing the original KCI statistic does not effectively improve its test power.
>
> Instead, we took a surrogate approach by decomposing the original KCI statistic and directly exposing the kernel parameters applied to the conditional variable for optimization.
> This partially improved the test power. While we cannot claim to have found the optimal kernel parameters as Gretton et al. (2012) did, this approach avoids the redundancy of re-learning the conditional mean embedding by selecting the kernel parameters on the response variable.
> In terms of results, our method achieves higher test power with minimal computational overhead and comparable Type I error control, which represents our improvement.
>
> [1] Gretton A, et al. Optimal kernel choice for large-scale two-sample tests[J]. NeurIPS, 2012.
>
> >**Q2.** It is unclear which parts are original contributions by the authors and which are drawn from existing literature. Please clarify this distinction, and provide appropriate references for any cited results.
>
> **A.** Thank you for your comments. In future revisions, we will more appropriately cite relevant papers and better highlight our contributions.
>
> >**Q3.** Does the proposed method learn only the kernel bandwidths, or does it also learn the kernel functions?
>
> **A.** Our method currently focuses on learning the parameters within a given kernel family (Gaussian kernel). Theoretically, our approach could also be extended to select between different kernel families (which might require further discussion on details such as whether the true power remains consistent for the same estimated power across different kernel families). However, as we stated in the paper, "This selection process primarily focuses on tuning kernel parameters, which can often be more influential than the choice of the kernel family itself." Different kernel families can exhibit similar filtering characteristics. Therefore, for now, we are concentrating on the former.
>
> >**Q4.** The asymptotic results are derived under fixed kernel parameters. How do the learned parameters affect these results? Moreover, how can the authors ensure that the proposed method at least asymptotically controls the type I error using the weighted sum of chi-squared to compute the p-value?
>
> **A.** Our method does not affect its asymptotic properties, regardless of whether the weighted sum of chi-squared or other approximation methods is used.
> The difference from the median heuristic-based one lies in how to determine kernel parameters involved before testing: the median heuristic uses a heuristic approach, while we propose determining them based on the estimated test power.
> During the testing phase, all kernel parameters in our method are fixed with the pre-split test samples.
> Therefore, this does not alter its asymptotic properties.
>
> >**Q5.** Could the authors provide a more detailed explanation of the impact of the conditional expectation bias?
>
> **A.** The misspecification of the conditional embedding can lead to residuals that still exhibit correlation induced by the conditioning variable.
> This correlation bias not only causes the shift in the mean of the statistic with limited samples (leading to uncontrollable Type I error) but also affects the estimation of test power.
> Considering this bias, we opted to fix the residual part.
> Although the bias remains, it is fixed. As the sample size increases, the estimated test power gradually converges, enabling the use of the estimated power for the selection of more appropriate kernel parameters.

---

> > ### Author Response · Authors · 2024-11-23
> >
> > >**Q1 & Q6.** Some phrases lack precision. For example, the notation of "feature mapping $\phi_x(x)$" and "$Z \sim \mathcal{N}(0, d_Z)$".
> >
> > **A.** Thank you for your thorough review. Here $ d_Z $ represents the dimension of $Z$, and we have already modified it and further checked our notation.
> >
> > >**Q7.** Does the original form of KCI in Section 4.1.2 use the median heuristic for determining kernel bandwidth?
> >
> > **A.** Yes, the original form of KCI uses its default parameter settings, where all kernel parameters are fixed using the median heuristic, and the conditioning $Z$ is not separated from the residual of $(X, Z)$ on $ Z$.
> >
> > >**Q8.** Does the discreteness of continuity of $X, Y, Z$ affect the performance of the proposed method, such as the choice of kernel?
> >
> > **A.** KCI is inherently designed for continuous data, so KCI-based methods, including ours, cannot be directly applied to analyze discrete data.
> > Due to the poor regression fitting results on discrete data, the learned residuals fail to block the influence of the conditioning variable on them, leading to poor performance.
> > Through experiments, we found that KCI performs poorly in controlling Type I error when handling discrete data, especially when the data is highly discretized with a small number of bins.
> > Therefore, we believe it cannot be directly applied to discrete data.

---

### Official Review · Reviewer_BJt5 · 2024-11-08

**Soundness:** 2
**Presentation:** 3
**Contribution:** 2
**Rating:** 5
**Confidence:** 3

**Summary:**

The submission describes an approach for data-driven kernel hyperparameters selection for improving the performance of kernel-based conditional independence hypothesis testing.
The approach is based on an asymptotic approximation of the test power and its optimization.

**Strengths:**

- very interesting topic and problem
- mostly clearly written.
- simulation with ablation study, time complexity and details.

**Weaknesses:**

1. the theoretical framework is somehow confusing to me. As stated by the authors "no method can simultaneously control the
Type I error rate at the given significance level while maintaining adequate power" citing Shah & Peters (2020). But then the authors reach the conclusion that "the practical evaluation of CI methods necessitates a balanced assessment of both Type I and Type
II error rates, emphasizing the trade-off between error control and statistical power". In my understanding this is not what we require from a statistical testing procedure, it is essential to control type I error, at least under some assumptions.
This is what Shah & Peters (2020) propose with the generalized covariance measure.
Of course, in practice, rarely assumptions are met, but if the statistical method proposed is not able to control type I error at least under some assumptions how it can be considered a valid testing procedure? I feel this is the difference from hypothesis testing and binary classification.
2. implications of 1. above are reflected in the evaluation in Figure 1, where it seems that only CIRCE is able to attain the required type I error control (of course with a higher type II error). Also about Figure 2 what is the meaning of testing CI with a 30 dimensional conditioning set with 1000 points without assumptions? It seems that is not possible and probably it should not be done no?

**Questions:**

see the weaknesses above and

-  An operator "is not Hilbert-Schmidt in characteristic RKHS" does it means that is not bounded ?, that is has not a bounded HS norm? this should be clarified in the text.

---

> ### Author Response · Authors · 2024-11-23
>
> We sincerely appreciate your profound and valuable suggestions for our paper. Thank you for your effort and dedication!
>
> >**W1.**  If the statistical method proposed is not able to control type I error, at least under some assumptions, how can it be considered a valid testing procedure?
>
> **A.**  Thank you for your insightful evaluation. The points regarding *"no method can simultaneously control the Type I error rate at the given significance level while maintaining adequate power"* and *"the practical evaluation of CI methods necessitates a balanced assessment of both Type I and Type II error rates, emphasizing the trade-off between error control and statistical power"* are grounded in a "practical" perspective.
>
> On one hand, CI testing inherently differs from other hypothesis testing tasks, such as two-sample tests or independence tests, because it specifically considers whether there is a shared latent confounder among observed variables. This involves testing the independence of latent variables, requiring the exposure of latent variables (which introduces misspecification bias due to limited samples in practice). Thus, without assumptions, no method can effectively control the Type I error while maintain valid test power when dealing with general CI cases in practice.
>
> On the other hand, while some approaches like GCM (Shah et al., 2020) and CRT (Candes et al., 2018) claim to effectively or asymptotically control Type I error, they often come with stringent conditions (e.g., assuming the conditional distribution is known or the in-sample mean-squared error is small), which is almost impossible to achieve or verify in practice.
>
> Therefore, we believe that in real-world tasks, an empirical evaluation for general CI testing still necessitates a comprehensive assessment from the perspectives of both *"error control and testing power"* in broad-ranging cases. This is also why we describe our approach as a *"practical kernel choice method."*
>
> >**W2.** On the Type I error performance in synthetic data and high-dimensional data.
>
> **A.** As mentioned in the response to Q1, KCI-based methods, whether CIRCE or our approach, cannot fully control Type I error. Therefore, from a practical perspective, it is still necessary to simultaneously compare both Type I error and Type II error.
>
> In high-dimensional experiments, we followed the experimental setting in Polo et al. (2023) to compare the performance of different methods under such extreme conditions. The results show that, even when all methods perform poorly, our approach still slightly outperforms the Median heuristic.
>
> We also followed another high-dimensional experiment from Li et al. (2024) We also followed another high-dimensional experiment from Li et al. (2024) (following the suggestion from Reviewer QG5Q). In this experiment, the conditioning set dimension was set to 100 , and the results are as follows. And KNN is the proposed method in Li et al. From the results, it can be observed that, even in high-dimensional scenarios, under certain specific settings, the kernel-based methods still demonstrate comparable Type I error control and test power.
>
> Type I Error with different sample size N ($\alpha = 0.05$)
>
> | **N** | CIRCE | Median| Power | KNN |
> |------------------|-----------|------------|-----------|---------|
> | 100              | 0         | 0.02       | 0.02      | 0.05    |
> | 300              | 0.08      | 0.04       | 0.04      | 0.04    |
> | 500              | 0.11      | 0.04       | 0.04      | 0.05    |
>
> Type II Error
>
> | **N** | **CIRCE** | **Median** | **Power** | **KNN** |
> |------------------|-----------|------------|-----------|---------|
> | 100              | 0.14      | 0.47       | 0.46      | 0.69    |
> | 300              | 0         | 0.03       | 0.01      | 0.26    |
> | 500              | 0         | 0          | 0         | 0.09    |
>
> >**Q1.** An operator "is not Hilbert-Schmidt in characteristic RKHS" does it means that is not bounded?
>
> **A.** Yes, because when $ \mathcal{H_Z} $ is infinite-dimensional, its Hilbert–Schmidt norm will be infinite, and thus it is not a Hilbert–Schmidt operator. Thank you for your suggestion; we will provide a more detailed explanation for this part.
>
> [1] Candes E, et al. Panning for gold:‘model-X’knockoffs for high dimensional controlled variable selection, 2018
>
> [2] Li et al. K-nearest-neighbor local sampling based conditional independence testing. 2024.

---

### Meta-Review · Area_Chair_qehM · 2024-12-17

**Metareview:**

**(a) Summary**

This paper investigates the kernel-based conditional independence (KCI) test, a widely used statistical approach for assessing conditional independence, and proposes a method for kernel parameter selection. The key technical contribution lies in providing a practical parameter selection method as an alternative to the well-established median heuristic. The effectiveness of the proposed method has been evaluated on synthetic and real-world datasets.

**(b) Strengths**

- **Motivation:** The paper is well-motivated. The KCI test is an important tool in machine learning and data analysis, and kernel parameter selection remains a challenging research problem. This is particularly crucial since parameter tuning in this context is inherently difficult, yet the resulting performance strongly depends on it.
- **Structure:** The paper is well-structured and easy to follow.
- **Mathematical discussion:** The mathematical descriptions are mostly clear and well-written.

**(c) Weaknesses**

- **Novelty:** There are concerns regarding the novelty of the contribution, as raised by several reviewers. While the authors provide detailed explanations in the rebuttal, these concerns remain. A study may still be valuable if it provides strong theoretical insights or demonstrates significant empirical improvements; however, this paper does not convincingly achieve either.
- **Presentation:** Several presentation issues remain, as noted by reviewers: The distinction between novel technical contributions and known results is unclear. The gap between theoretical assumptions and practical scenarios is insufficiently discussed. Many sections require more careful explanations to improve clarity. While the rebuttal addressed some of these concerns, a major revision is needed to refine the presentation.
- **Empirical performance:** The empirical results are not particularly promising. The proposed method shows only marginal improvements over the median heuristic, and its advantages are not clearly demonstrated.

**(d) Decision Reasoning**

After carefully reviewing the paper and the rebuttal, I conclude that the weaknesses outlined above must be addressed before publication. While the technical contribution is potentially interesting, the most critical concern is the empirical performance. The proposed method offers only marginal gains over the median heuristic, which raises questions about its practical utility. A more thorough discussion and clear demonstration of scenarios where the proposed approach outperforms the median heuristic are necessary.

**Additional Comments On Reviewer Discussion:**

Crucial concerns regarding the novelty, empirical performance, and presentation were raised by the reviewers. While I believe the authors have mostly addressed these concerns in their rebuttal, the necessary revisions to incorporate the rebuttal feedback have not been submitted during the discussion period. As a result, I conclude that the paper requires at least one more round of revisions. Therefore, I recommend rejecting the paper at this stage.

---

### Decision · Program_Chairs · 2025-01-22

Reject